# Model Monotonicity in Autobidding Auctions:
# When Do Better Predictions Lead to Better Outcomes?

**Ashwinkumar Badanidiyuru** [1]

## Abstract

Online advertising platforms rely on machine learning models to predict click-through rates (pCTR) and conversion rates (pCVR) for auction mechanisms. We introduce a novel framework to study the interaction between recommender system model quality, auction format, and autobidder behavior. We formalize when model improvements—defined via a refinement relation inspired by filtrations in probability theory—lead to improvements in platform-level Evaluation Criteria Metrics (ECM) such as revenue, welfare, or liquid welfare. Our main contributions are: (1) a formal definition of model improvement based on cluster refinement, and (2) a systematic characterization of ECM monotonicity across different combinations of bidder types (tCPA, max-CPA), auction formats (first-price, second-price, VCG), and budget constraints. We show that first-price auctions with uniform bidding guarantee revenue monotonicity for tCPA bidders without budgets (via Jensen's inequality), while second-price auctions and budget constraints can break this property. We provide full numerical constructions for the non-monotonicity results. Our findings have practical implications for advertising platforms seeking to align model improvements with business outcomes.

## 1. Introduction

Online advertising platforms generate hundreds of billions of dollars in annual revenue by matching advertisers with users through sophisticated auction mechanisms. At the heart of these systems lies a critical interplay between three components: (1) machine learning models that predict user behavior, such as click-through rate (pCTR) and conversion rate (pCVR) models; (2) auction mechanisms that allocate ad impressions and determine prices; and (3) autobidding systems that bid on behalf of advertisers according to their specified objectives. Understanding how improvements in one component propagate through this system to affect platform-level outcomes is essential for both theoretical understanding and practical system design.

Advertising platforms continually invest in improving their prediction models, with the natural expectation that better predictions lead to better outcomes. However, the relationship between model quality and platform metrics is far from straightforward. When a platform deploys a more accurate pCTR model, the change affects not only the auction mechanism's allocation decisions but also the behavior of autobidders, which adapt their bidding strategies to the new predictions. This creates a complex feedback loop where the ultimate effect on platform metrics—such as revenue, advertiser welfare, or efficiency—depends on the intricate interaction between all three components.

### 1.1. Motivation and Central Question

Consider a platform that improves its conversion prediction model, enabling finer-grained distinctions between users. Intuitively, one might expect that more accurate predictions should lead to more efficient allocations and higher revenue. Yet this intuition can fail. A refined model may enable some autobidders to target only the most valuable impressions more precisely, potentially reducing the competitive pressure that drives revenue. Alternatively, budget-constrained advertisers might exhaust their budgets more quickly on high-value users, leaving lower-value users unserved. These scenarios illustrate that the relationship between model quality and platform outcomes is subtle and depends critically on the auction format, bidder objectives, and resource constraints.

This motivates our central research question: *When do improvements in prediction models lead to improvements in platform-level Evaluation Criteria Metrics (ECM)?* We refer to this property as **ECM monotonicity** or simply **model monotonicity**. Understanding when this property holds—and when it fails—has significant implications for platform

---

[1]Uber Technologies, Inc., San Francisco, CA, USA. Correspondence to: Ashwinkumar Badanidiyuru <ashwinkumarbv@uber.com>.

*Proceedings of the 43rd International Conference on Machine Learning*, Seoul, South Korea. PMLR 306, 2026. Copyright 2026 by the author(s).

design, model deployment decisions, and the alignment of technical improvements with business objectives.

## 1.2. Our Contributions

We make two main contributions in this paper:

**Novel Problem Formulation.** We introduce a formal framework for studying the interaction between model quality, auction mechanisms, and autobidder behavior. Central to our approach is a definition of *model refinement* inspired by filtrations in probability theory. We model prediction systems as partitioning a universe $\mathcal{U}$ of users into clusters, where the model predicts the average conversion probability for each cluster. A model $\mathcal{M}_A$ *refines* a model $\mathcal{M}_B$ if each cluster of $\mathcal{M}_A$ is contained in a cluster of $\mathcal{M}_B$—that is, $\mathcal{M}_A$ makes finer distinctions between users. This definition captures the intuitive notion that $\mathcal{M}_A$ provides "more information" than $\mathcal{M}_B$, while being mathematically tractable for analysis.

Within this framework, we consider two types of autobidders that reflect common industry practice:

- **Target CPA (tCPA) bidders**: Advertisers specify a target cost-per-acquisition and seek to maximize conversion volume subject to maintaining an average CPA at or below their target.

- **MAX-CPA bidders**: Advertisers specify a maximum willingness-to-pay per conversion and seek to maximize total value, where each conversion's value equals the advertiser's maximum CPA.

Following standard autobidding practice (Aggarwal et al., 2024), we assume tCPA targets equal true per-conversion values ($v_i = t_i$), ensuring welfare equals revenue for tCPA bidders without budgets under FPA with multiplier $\mu_i = 1$. We study these bidder types across multiple auction formats (first-price, VCG—which is equivalent to second-price for single-item auctions) and under both unconstrained and budget-constrained settings.

**Systematic Characterization of ECM Monotonicity.** We provide a comprehensive analysis of when model refinement leads to improvements in platform-level metrics, with our systematic characterization in Table 1. We find that monotonicity is *rare*—only three settings guarantee it: (1) tCPA bidders without budgets under FPA (via Jensen's inequality), (2) MAX-CPA bidders without budgets under VCG (welfare only), and (3) LP-based fractional allocation with budgets (welfare monotonic; for tCPA, surrogate revenue also monotonic; not incentive compatible). In all remaining settings studied—including VCG for tCPA bidders and FPA with budget constraints—monotonicity fails.

We provide explicit numerical constructions for these non-monotonicity results.

**Key Insights.** Our results yield several insights: (1) Monotonicity is rare—only three settings guarantee it, so platforms should not assume model improvements translate to metric improvements. (2) FPA provides the only revenue monotonicity guarantee (for tCPA without budgets), while VCG provides the only welfare guarantee (for MAX-CPA without budgets). (3) A fundamental trade-off exists between incentive compatibility and monotonicity with budget constraints. (4) Model improvements should be validated through simulation or A/B testing.

**Conflict of Interest Disclosure.** The author is employed by Uber, which operates advertising systems that use auction mechanisms.

## 2. Related Work

Our work studies model monotonicity in autobidding auctions, connecting several research areas. We briefly survey the most relevant literature, referring readers to standard references for foundational concepts.

**Autobidding.** The theoretical study of autobidding was initiated by Aggarwal et al. (2019), who analyzed constrained autobidding and established foundational equilibrium concepts. Aggarwal et al. (2024) provide a comprehensive survey covering equilibrium analysis, mechanism design, and learning in autobidding systems. Recent work extends to multi-channel settings (Deng et al., 2023) and dynamics with budget and ROI constraints (Lucier et al., 2024). These works analyze autobidder behavior given fixed prediction models; our work studies how changes in model quality affect equilibrium outcomes.

**First-Price Auctions and Pacing.** The shift to first-price auctions has motivated substantial work on pacing equilibria (Conitzer et al., 2022) and efficiency in non-truthful settings (Liaw et al., 2023; 2024). Deng et al. (2024) empirically study bid-scaling strategies across auction formats. Our positive result for tCPA bidders under FPA connects to this literature, while our negative results highlight limitations of first-price mechanisms.

**VCG and Welfare.** The VCG mechanism (Vickrey, 1961; Clarke, 1971; Groves, 1973) achieves welfare-optimal allocations with dominant-strategy incentive compatibility. Our welfare monotonicity result for MAX-CPA bidders under VCG builds on these classical efficiency properties. Mehta (2022) shows that randomization can improve efficiency beyond VCG in autobidding contexts.

**Value Maximizers and Liquid Welfare.** Max-CPA bidders are a form of value maximizers (Wilkens et al., 2016). For budget-constrained settings, liquid welfare (Dobzinski & Paes Leme, 2014) provides an appropriate efficiency benchmark; Colini-Baldeschi et al. (2026) study optimal liquid welfare guarantees for autobidding agents with budgets. Our negative results for budget-constrained settings show that even liquid welfare fails to be monotonic.

**Prediction Models in Advertising.** Vasile et al. (2017) observe that prediction accuracy may not align with business objectives due to downstream auction effects—a theme central to our work. While prior work focuses on learning better predictions or bidding strategies (Ma et al., 2018; Cai et al., 2017), we study when improving predictions actually improves platform outcomes.

**Positioning.** To our knowledge, this is the first work to formally study when prediction model improvements guarantee platform metric improvements in autobidding settings. Our key finding—that monotonicity holds only in a small set of specific cases—provides theoretical foundations for understanding the complex dynamics of model deployment in advertising platforms.

## 3. Model Definition and Problem Setting

We formalize the problem setting for studying the interaction between prediction model quality and platform-level outcomes in autobidding auctions. Our framework draws on concepts from probability theory, specifically filtrations and refinements (Hardy et al., 1952; Durrett, 2019).

### 3.1. Problem Setting

We consider an online advertising platform that hosts auctions to display advertisements to users. Let $\mathcal{U}$ denote the *user universe*, a finite set of users. Each user $u \in \mathcal{U}$ and advertiser $i \in \mathcal{A}$ are associated with true (unobservable) behavioral parameters $\mathrm{ctr}_i(u), \mathrm{cvr}_i(u) \in [0,1]$ representing click-through and conversion rates for that advertiser–user pair.

A set of advertisers $\mathcal{A} = \{1, \dots, n\}$ compete for ad slots, each with private value $v_i \in \mathbb{R}_{\geq 0}$ per conversion and constraints such as target CPA (tCPA) or maximum CPA. Advertisers employ *autobidding* systems (Aggarwal et al., 2019; 2024) that automatically determine bids based on the platform's predicted values $\mathrm{pCTR}(u)$ and $\mathrm{pCVR}(u)$.

**Assumption 3.1** (Value Equals Target for tCPA Bidders). For target CPA (tCPA) bidders, we assume the per-conversion value $v_i$ equals the target $t_i$. This ensures that, without budget constraints and under first-price auctions with multiplier $\mu_i = 1$, welfare equals revenue for tCPA bidders, since the expected value $v_i \cdot p_{i,C}$ equals the target

payment $t_i \cdot p_{i,C}$.

### 3.2. Prediction Models

We formalize prediction models using a partition-based framework that captures prediction granularity.

**Definition 3.2** (User Partition). A *partition* of $\mathcal{U}$ is a collection $\mathcal{P} = \{C_1, \dots, C_k\}$ of non-empty, pairwise disjoint subsets with $\bigcup_{j=1}^{k} C_j = \mathcal{U}$. For each cluster $C \in \mathcal{P}$, we define its *weight* $w_C := |C|/|\mathcal{U}|$, representing the probability mass (or relative size) of cluster $C$ under the uniform user distribution.

**Definition 3.3** (Prediction Model). A *prediction model* $\mathcal{M}$ is defined by a partition $\mathcal{P}_{\mathcal{M}}$ of $\mathcal{U}$. For each advertiser $i \in \mathcal{A}$ and cluster $C \in \mathcal{P}_{\mathcal{M}}$, the model outputs the *predicted conversion probability*:

$$p_{i,C} := \frac{1}{|C|} \sum_{u \in C} p_i(u), \tag{1}$$

where $p_i(u) = \mathrm{ctr}_i(u) \cdot \mathrm{cvr}_i(u)$ is the true conversion probability for advertiser $i$ and user $u$. This quantity $p_{i,C}$ is the primitive used in all auction analysis: bidders bid based on $p_{i,C}$, and all theoretical results in this paper are stated in terms of $p_{i,C}$.

This definition captures the intuition that a prediction model groups users into clusters and assigns the same predicted conversion probability to all users within a cluster. The model is *calibrated*: predictions equal true cluster averages (Ma et al., 2018; Gneiting & Raftery, 2007). Crucially, this paper studies the effect of *refining a perfectly calibrated information structure* rather than the effect of stochastic prediction error or miscalibration. (See Appendix A.1 for implementation details.)

**Example 3.4** (Extreme Models). The *coarsest model* $\mathcal{P} = \{\mathcal{U}\}$ predicts a single global average: $p_{i,\mathcal{U}} = \frac{1}{|\mathcal{U}|} \sum_{u \in \mathcal{U}} p_i(u)$ for all advertisers $i$. The *finest model* $\mathcal{P} = \{\{u\} : u \in \mathcal{U}\}$ predicts exact advertiser-specific conversion probabilities: $p_{i,\{u\}} = p_i(u)$ for each user $u$ and advertiser $i$.

### 3.3. Model Refinement

We define when one prediction model is "better" than another, inspired by filtrations in probability theory (Durrett, 2019). Unlike pointwise metrics (log-likelihood, AUC), model refinement provides a partial order under which proper scoring losses improve in expectation for calibrated nested partitions, enabling provable guarantees.

**Definition 3.5** (Model Refinement). A prediction model $\mathcal{M}_A$ *refines* model $\mathcal{M}_B$ (written $\mathcal{M}_A \succeq \mathcal{M}_B$) if every cluster in $\mathcal{P}_{\mathcal{M}_A}$ is contained in some cluster of $\mathcal{P}_{\mathcal{M}_B}$. We say that $\mathcal{M}_A$ is *finer* and $\mathcal{M}_B$ is *coarser*.

This captures feature-set refinements: for example, a model segmenting users by {age, location} induces a coarser partition than a model that additionally uses {browsing history}, provided the added feature only splits existing clusters.

The refinement relation forms a partial order (reflexivity, transitivity, antisymmetry follow from standard set theory). In probability theory, a finer partition corresponds to a finer $\sigma$-algebra, capturing the idea that a finer model has access to more discriminative information.

The refinement definition captures a strong notion of model improvement: if $\mathcal{M}_A \succeq \mathcal{M}_B$, then for any advertiser $i$ and any concave Bayes risk $H$ associated with a proper scoring loss,

$$\sum_{C \in \mathcal{P}_{\mathcal{M}_A}} w_C H(p_{i,C}) \leq \sum_{C \in \mathcal{P}_{\mathcal{M}_B}} w_C H(p_{i,C}).$$

This follows from Jensen's inequality because each coarse prediction is the weighted average of the refined predictions within that coarse cluster. Thus refinement improves all proper scoring losses in their loss orientation (Gneiting & Raftery, 2007).

### 3.4. Evaluation Criteria Metric (ECM)

**Definition 3.6** (Evaluation Criteria Metric). An *Evaluation Criteria Metric* (ECM) aggregates auction outcomes. For a model $\mathcal{M}$, let $x_i(u; \mathcal{M}) \in \{0, 1\}$ denote the allocation and $\pi_i(u; \mathcal{M}) \geq 0$ the payment for advertiser $i$ on user $u$. We consider:

- **Revenue:** $\mathrm{Revenue}(\mathcal{M}) := \sum_{u \in \mathcal{U}} \sum_{i \in \mathcal{A}} \pi_i(u; \mathcal{M})$

- **Welfare:** $\mathrm{Welfare}(\mathcal{M}) := \sum_{u \in \mathcal{U}} \sum_{i \in \mathcal{A}} x_i(u; \mathcal{M}) \cdot v_i \cdot p_{i,u}$

- **Liquid Welfare:** $\mathrm{LiquidWelfare}(\mathcal{M}) := \sum_{i \in \mathcal{A}} \min\left(B_i, \sum_u x_i(u; \mathcal{M}) \cdot v_i \cdot p_{i,u}\right)$

Here $p_{i,u}$ denotes the true conversion probability for advertiser $i$ and user $u$, and $B_i$ is advertiser $i$'s budget (Dobzinski & Paes Leme, 2014). In proofs, we write metrics as weighted sums over clusters (see Appendix A.1).

For each setting, let $\Phi_{\mathrm{setting}}(\mathcal{M})$ denote the outcome mapping that selects allocations and payments induced by model $\mathcal{M}$ under that setting's behavioral convention.

**Definition 3.7** (ECM Monotonicity). An ECM is *monotone* with respect to model refinement for a given auction format and bidder type if, for all models $\mathcal{M}_A, \mathcal{M}_B$ with $\mathcal{M}_A \succeq \mathcal{M}_B$: $\mathrm{ECM}(\Phi_{\mathrm{setting}}(\mathcal{M}_A)) \geq \mathrm{ECM}(\Phi_{\mathrm{setting}}(\mathcal{M}_B))$, where the ECM is evaluated on the allocations and payments selected by the outcome mapping for that setting (see Assumption 4.10).

*Remark* 3.8 (Setting-Dependent Outcome Mappings). The outcome mapping $\Phi_{\mathrm{setting}} : \mathcal{M} \to$ (allocations, payments) varies by setting. For tCPA settings, $\Phi$ selects equilibrium outcomes under the binding-constraint convention. For MAX-CPA FPA, $\Phi$ compares designated feasible multiplier profiles across models without claiming a full equilibrium characterization. Our characterization in Table 1 is thus a taxonomy of $\Phi$-monotonicity results across these conventions. This heterogeneity reflects the different behavioral assumptions appropriate to each setting.

### 3.5. Central Question

We study which combinations of (auction format, bidder type, ECM) exhibit monotonicity:

- **Auction formats:** First-price (FPA), VCG mechanism (equivalent to SPA for single-item auctions).

- **Bidder types:** Target CPA (tCPA) and Maximum CPA (max-CPA).

- **Budget constraints:** Present or absent.

*Central Question:* For which combinations is the ECM monotone with respect to model refinement?

A positive answer ensures that better models lead to better outcomes. A negative answer reveals misalignment: improving predictions can paradoxically hurt platform objectives. We provide a systematic characterization in Section 5, with proofs for positive results and numerical constructions for non-monotonicity results.

## 4. Auction Mechanisms and Bidder Models

We formalize the auction mechanisms and bidder models underlying our analysis. For comprehensive background on auction theory, see Krishna (2009); for autobidding systems, see Aggarwal et al. (2019).

### 4.1. Auction Environment

Consider advertisers $\mathcal{A} = \{1, \ldots, n\}$ competing for impressions across a user population partitioned into clusters $\mathcal{P}$ by prediction model $\mathcal{M}$. For each advertiser $i$ and cluster $C$, the platform observes predicted conversion probability $p_{i,C}$, defined as the cluster-averaged conversion probability $p_{i,C}$ (Definition 3.3). We assume the model is well-calibrated within each cluster.

### 4.2. Bidder Models

**Definition 4.1** (tCPA Bidder). A *target cost-per-acquisition (tCPA)* bidder $i$ has target $t_i > 0$ and seeks to maximize conversions subject to average cost per conversion $\leq t_i$.

**Definition 4.2** (MAX-CPA Bidder). A *maximum cost-per-acquisition (MAX-CPA)* bidder $i$ has per-conversion value $v_i > 0$ and maximizes total expected value (conversions times $v_i$) minus total payment. This bidder obtains positive surplus from an impression only when expected cost per conversion satisfies $\pi/p_{i,C} \leq v_i$, where $\pi$ denotes the per-impression payment.

The key distinction: tCPA bidders maximize conversions subject to an average cost constraint; MAX-CPA bidders maximize quasi-linear utility (value minus payment). Either type may have a budget constraint $B_i > 0$ limiting total expenditure.

### 4.3. Uniform Bidding

**Definition 4.3** (Uniform Bidding). Under *uniform bidding*, bidder $i$ selects multiplier $\mu_i \geq 0$ and bids $b_i(C) = \mu_i \cdot t_i \cdot p_{i,C}$ (tCPA) or $b_i(C) = \mu_i \cdot v_i \cdot p_{i,C}$ (MAX-CPA).

For MAX-CPA bidders, we define the *effective multiplier* $g_i := \mu_i \cdot v_i$, so bids simplify to $b_i(C) = g_i \cdot p_{i,C}$.

**Lemma 4.4** (Optimal Multiplier for tCPA in FPA). *For a tCPA bidder without budget constraints in a first-price auction, the optimal uniform multiplier is $\mu_i^* = 1$.*

*Proof Sketch.* In FPA with uniform bidding $b_i(C) = \mu_i \cdot t_i \cdot p_{i,C}$, the CPA in any won cluster is $\mu_i \cdot t_i$. Thus $\mu_i \leq 1$ is required for feasibility. Since conversions increase with $\mu_i$, the optimal choice is $\mu_i = 1$. $\square$

### 4.4. Auction Formats

**Definition 4.5** (First-Price Auction (FPA)). The highest bidder wins and pays their bid: $i^* = \arg\max_i b_i$, payment $\pi_{i^*} = b_{i^*}$. (Ties broken by lowest index.)

**Definition 4.6** (Second-Price Auction (SPA)). The highest bidder wins and pays the second-highest bid: $i^* = \arg\max_i b_i$, payment $\pi_{i^*} = \max_{j \neq i^*} b_j$.

*Remark* 4.7 (SPA-VCG Equivalence). For single-item auctions, SPA and VCG are equivalent mechanisms: both allocate to the highest bidder and charge the second-highest bid. The VCG payment equals the externality imposed on other bidders, which for a single item is precisely the second-highest bid. Hence, all results stated for VCG in single-item settings apply equally to SPA.

**Definition 4.8** (VCG Mechanism). VCG computes the welfare-maximizing allocation and charges each winner their externality (the welfare loss imposed on others).

**Definition 4.9** (LP Allocation Benchmark). The *LP allocation benchmark* is a centralized fractional allocation-and-payment rule that directly chooses allocation variables $x_{i,C} \in [0,1]$ subject to supply and budget feasibility constraints. Unlike FPA, SPA, or VCG, this benchmark is not a strategic auction mechanism: it assumes the platform knows the relevant bidder parameters and optimizes allocations and surrogate payments directly. We use it only as a comparison point for monotonicity under budget constraints.

### 4.5. Evaluation Criteria Metrics

We use the revenue, welfare, and liquid welfare metrics from Definition 3.6. In the auction analysis, welfare is written cluster-wise as $\sum_i \mathbb{E}[\mathbf{1}[i \text{ wins}] \cdot v_i \cdot p_{i,C}]$, and liquid welfare caps each bidder's contribution at their budget.

### 4.6. Equilibrium Concept

**Assumption 4.10** (Equilibrium Convention). Bidders choose uniform multipliers $\mu_i$ to optimize their objective (conversions subject to CPA $\leq t_i$ for tCPA; quasi-linear utility for MAX-CPA) taking others' multipliers as given. We assume an equilibrium exists. When multiple equilibria preserve the same allocation, our positive results hold for all such equilibria; counterexamples exhibit one valid equilibrium (typically with binding CPA constraints for transparency), except for MAX-CPA FPA which uses designated feasible multiplier profiles without a full equilibrium claim. For FPA with tCPA bidders, symmetric equilibrium corresponds to $\mu_i = 1$ (truthful bidding).

## 5. Main Results

This section presents our main theoretical contributions: a systematic characterization of when model refinement leads to improved platform metrics. We state all positive results with proof sketches (full proofs in Appendix A) and reference the detailed numerical constructions in Section 6 for non-monotonicity results.

### 5.1. Summary of Results

Our main finding is that ECM monotonicity depends critically on the interaction between three factors: (1) bidder type (tCPA vs. MAX-CPA), (2) auction format (FPA, SPA, VCG), and (3) presence of budget constraints. Table 1 summarizes our systematic characterization. The table reports the auction and benchmark settings analyzed in this paper, rather than an exhaustive enumeration of every possible budgeted mechanism variant. Note that monotonicity is defined relative to a setting-dependent outcome mapping $\Phi_{\text{setting}}$; see Remark 3.8 for details on how outcomes are determined in each row.

---

[1]For unconstrained settings, welfare is standard social welfare. For budget-constrained settings, welfare refers to liquid welfare (Definition 3.6), which caps each bidder's contribution at their budget. *Outcome selection:* For tCPA auction settings (rows 1–3), ECM is evaluated at equilibrium multipliers where the binding constraint (CPA or budget) determines each bidder's multiplier;

*Table 1.* Monotonicity of ECM under model refinement. ✓indicates monotonicity holds; × indicates a non-monotonicity construction exists; — indicates metric not defined for this setting. Results assume uniform bidding. For budget-constrained settings, welfare refers to liquid welfare.[1]

| Bidder Type | Auction | Revenue | Welfare |
|---|---|---|---|
| tCPA | FPA | ✓ | ✓ |
| tCPA | VCG (= SPA) | × | × |
| tCPA + Budget | FPA | × | × |
| tCPA + Budget | LP | ✓ | ✓ |
| MAX-CPA | FPA$^\dagger$ | × | × |
| MAX-CPA | VCG | × | ✓ |
| MAX-CPA + Budget | FPA$^\dagger$ | × | × |
| MAX-CPA + Budget | LP | — | ✓ |

## 5.2. Positive Results: Monotonicity Guarantees

We begin with our positive results, establishing conditions under which model refinement provably improves platform metrics.

### 5.2.1. FIRST-PRICE AUCTION WITH tCPA BIDDERS

Our first main result shows that FPA with tCPA bidders and uniform bidding satisfies revenue monotonicity.

**Theorem 5.1** (FPA Revenue Monotonicity). *Consider tCPA bidders without budget constraints using uniform bidding with optimal multiplier $\mu_i = 1$ in a first-price auction. If model $\mathcal{M}_A$ is a refinement of model $\mathcal{M}_B$ (i.e., $\mathcal{M}_A \succeq \mathcal{M}_B$), then:*

$$Rev(\mathcal{M}_A) \geq Rev(\mathcal{M}_B). \tag{2}$$

*Proof Sketch.* With $\mu_i = 1$, bidder $i$ bids $b_i(C) = t_i \cdot p_{i,C}$, so revenue is $\text{Rev}(\mathcal{M}) = \sum_{C \in \mathcal{P}} w_C \cdot \max_i t_i \cdot p_{i,C}$. The function $f(p) = \max_i t_i \cdot p_i$ is convex. When $\mathcal{M}_A$ refines $\mathcal{M}_B$, each coarse cluster $C^B$ splits into sub-clusters with weights summing to $w_{C^B}$ and calibration-preserving probabilities. By Jensen's inequality applied to this convex $f$, the finer model's revenue cannot be smaller. See Appendix A for the complete proof. $\square$

The following theorem establishes that uniform bidding with $\mu = 1$ is optimal for tCPA bidders in FPA who seek to maximize conversions subject to their tCPA constraint.

**Theorem 5.2** (FPA Optimality for tCPA Bidders). *For tCPA bidders without budget constraints in a first-price auction, uniform bidding with multiplier $\mu_i = 1$ (i.e., bidding $b_i = t_i \cdot p_{i,C}$) is:*

when unconstrained, $\mu = 1$ is used. The dagger marks the MAX-CPA FPA rows (rows 5 and 7), where ECM compares designated feasible multiplier profiles across models without a full equilibrium claim (see Assumption 4.10); for MAX-CPA VCG (row 6), equilibrium multipliers are used. For tCPA + Budget LP, revenue refers to surrogate first-price payments under the LP allocation.

1. *Optimal given the tCPA constraint: maximizes conversions while satisfying the target CPA.*

2. *Revenue-maximizing among uniform bidding equilibria: lower multipliers result in lower bids and lower revenue.*

3. *Welfare-maximizing: achieves the maximum possible social welfare.*

*Proof Sketch.* **(1)** With $b_i(C) = \mu_i \cdot t_i \cdot p_{i,C}$, the CPA per won cluster is $\mu_i \cdot t_i$, so $\mu_i \leq 1$ is required. Since conversions increase with $\mu_i$, optimality requires $\mu_i = 1$. **(2)** At $\mu = 1$, bids are maximal under the CPA constraint. **(3)** Allocation goes to $\arg\max_i t_i \cdot p_{i,C}$, which is welfare-maximizing since value equals $t_i \cdot p_{i,C}$. $\square$

**Corollary 5.3** (FPA Welfare Monotonicity for tCPA). *Under the conditions of Theorem 5.1 and Assumption 3.1 ($v_i = t_i$), with the optimal FPA multiplier $\mu_i = 1$, welfare is also monotone: Welfare($\mathcal{M}_A$) $\geq$ Welfare($\mathcal{M}_B$).*

*Proof.* For tCPA bidders bidding $b_i = t_i \cdot p_{i,C}$ with $v_i = t_i$, revenue equals welfare (the winner pays exactly their expected value $v_i p_{i,C}$). Thus welfare monotonicity follows directly from revenue monotonicity. $\square$

*Remark* 5.4 (Incentive Compatibility in Autobidding (IC-ROS)). FPA with uniform bidding is *incentive compatible with respect to return-on-spend constraints* (IC-ROS) in the terminology of Aggarwal et al. (2024). This autobidding-specific notion of IC means that, when tCPA targets $t_i$ and budgets $B_i$ are private information but predicted conversion probabilities $p_{i,C}$ are public, each bidder's optimal strategy is to truthfully report their constraints to the autobidding system. The autobidder then bids $b_i = \mu_i \cdot t_i \cdot p_{i,C}$, maximizing conversions subject to the reported constraints. This differs from classical dominant-strategy incentive compatibility (DSIC), which applies to quasi-linear utilities; tCPA bidders have non-linear objectives (conversion maximization subject to cost constraints).

### 5.2.2. VCG MECHANISM WITH MAX-CPA BIDDERS

The VCG mechanism provides welfare monotonicity guarantees specifically for MAX-CPA bidders.

**Theorem 5.5** (VCG Welfare Monotonicity for MAX-CPA). *Consider MAX-CPA bidders without budget constraints. Under the VCG mechanism, if $\mathcal{M}_A \succeq \mathcal{M}_B$, then:*

$$Welfare(\mathcal{M}_A) \geq Welfare(\mathcal{M}_B). \tag{3}$$

*Proof Sketch.* VCG allocates to maximize welfare, and truthful bidding ($\mu_i = 1$) is dominant for MAX-CPA bidders. Welfare is $\text{Welfare}(\mathcal{M}) = \sum_{C \in \mathcal{P}} w_C \cdot \max_i v_i \cdot p_{i,C}$,

which has identical structure to FPA revenue with $v_i$ replacing $t_i$. By the same Jensen argument (since $\max_i v_i \cdot p_i$ is convex), refinement cannot decrease welfare. $\square$

*Remark* 5.6 (VCG Revenue Non-Monotonicity). VCG revenue monotonicity does *not* hold for MAX-CPA bidders. VCG payments depend on the externality imposed on other bidders, which can increase or decrease under model refinement. See Appendix B.3 for a counterexample.

*Remark* 5.7 (VCG with tCPA Bidders). For tCPA bidders, VCG provides monotonicity for *neither* revenue nor welfare. The issue is that tCPA bidders do not have fixed per-conversion values—their effective value depends on the tCPA constraint binding status across clusters. Model refinement can change which clusters are binding, leading to non-monotonic outcomes. See Appendix B.1 for a counterexample.

## 5.3. Negative Results: Non-Monotonicity

We now state our negative results. Detailed numerical constructions are provided in Section 6, with full calculations in Appendix B.

### 5.3.1. VCG/SPA NON-MONOTONICITY FOR TCPA

**Theorem 5.8** (VCG/SPA Non-Monotonicity for tCPA). *There exist instances with tCPA bidders where VCG (equivalently, SPA for single-item auctions) exhibits non-monotonicity in both revenue and welfare:*

$$\mathcal{M}_A \succeq \mathcal{M}_B \quad but \quad Rev(\mathcal{M}_A) < Rev(\mathcal{M}_B), \quad (4)$$
$$\mathcal{M}_A \succeq \mathcal{M}_B \quad but \quad Welfare(\mathcal{M}_A) < Welfare(\mathcal{M}_B). \quad (5)$$

The key intuition for SPA/VCG non-monotonicity is that payments depend on the *second-highest* bid (the externality). Model refinement can cause the ranking of bidders to change across sub-clusters in ways that reduce competitive pressure, thereby reducing payments. For tCPA bidders specifically, the mismatch between tCPA constraints and VCG's welfare-maximizing allocation compounds this effect. See Appendix B.1 for a counterexample demonstrating a 6.2% decrease in both revenue and welfare.

**Theorem 5.9** (VCG Revenue Non-Monotonicity for MAX-CPA). *There exist instances with MAX-CPA bidders where VCG welfare is monotone but revenue is not:*

$$\mathcal{M}_A \succeq \mathcal{M}_B \quad but \quad Rev(\mathcal{M}_A) < Rev(\mathcal{M}_B). \quad (6)$$

VCG payments are externality-based: each winner pays the welfare loss they impose on others. Model refinement can reduce competition in certain clusters, decreasing externalities and thus payments. See Appendix B.3 for a counterexample.

### 5.3.2. BUDGET CONSTRAINTS BREAK MONOTONICITY

**Theorem 5.10** (FPA with Budget Non-Monotonicity). *There exist instances with tCPA bidders having budget constraints where model refinement decreases FPA revenue:*

$$\mathcal{M}_A \succeq \mathcal{M}_B \quad but \quad Rev(\mathcal{M}_A) < Rev(\mathcal{M}_B). \quad (7)$$

With budgets, the optimal multiplier is no longer $\mu = 1$. Bidders must pace to avoid depleting their budget too quickly. Model refinement can change competitive dynamics—in particular, revealing segments where a budget-constrained bidder can win more auctions with lower bids. See Appendix B.2 for a numerical counterexample demonstrating a 16.8% revenue decrease.

### 5.3.3. LP MONOTONICITY WITH BUDGET CONSTRAINTS

**Theorem 5.11** (LP Benchmark Monotonicity). *Consider a centralized LP benchmark that chooses fractional allocations and surrogate payments directly, rather than modeling strategic bidder behavior in an auction. The LP explicitly enforces budget feasibility through linear constraints of the form $\sum_C w_C x_{i,C} a_i p_{i,C} \leq B_i$, where $a_i$ is a bidder-specific constant (e.g., $t_i$ for tCPA surrogate payments or $v_i$ for MAX-CPA value budgets). This LP-based fractional allocation benchmark guarantees welfare monotonicity. Define:*

$$Welfare_{LP}(\mathcal{M}) := \sum_{i \in \mathcal{A}} \sum_{C \in \mathcal{P}} w_C \cdot x_{i,C}^* \cdot v_i \cdot p_{i,C}, \quad (8)$$

*where $x^* = \{x_{i,C}^*\}$ is the optimal LP solution. For tCPA bidders (where $v_i = t_i$), we additionally define:*

$$Rev_{LP}(\mathcal{M}) := \sum_{i \in \mathcal{A}} \sum_{C \in \mathcal{P}} w_C \cdot x_{i,C}^* \cdot t_i \cdot p_{i,C}. \quad (9)$$

*Note: $Rev_{LP}$ is a surrogate metric representing payments assigned by the LP benchmark, not equilibrium revenue from a strategic auction (see Remark 5.12). For tCPA bidders at $\mu_i = 1$, these surrogate payments coincide with first-price payments and $Rev_{LP} = Welfare_{LP}$ since $v_i = t_i$.*

*If $\mathcal{M}_A \succeq \mathcal{M}_B$, then:*

- *For tCPA or MAX-CPA bidders: $Welfare_{LP}(\mathcal{M}_A) \geq Welfare_{LP}(\mathcal{M}_B)$.*

- *For tCPA bidders only: $Rev_{LP}(\mathcal{M}_A) \geq Rev_{LP}(\mathcal{M}_B)$.*

*Proof Sketch.* A *lifting* argument: given optimal $x^B$ for $\mathcal{M}_B$, construct feasible $x^A$ for $\mathcal{M}_A$ by copying each allocation to all sub-clusters ($x_{i,C'}^A = x_{i,C}^B$ for $C' \subseteq C$). Calibration preservation ensures supply, budget, and objective all match. Thus the refined optimum cannot be smaller. See Appendix B.5. $\square$

*Remark* 5.12 (LP as a Non-Strategic Benchmark). Unlike FPA with uniform bidding (Remark 5.4), LP-based allocation is *not* incentive compatible and should be interpreted as a centralized allocation-and-payment benchmark, not an equilibrium statement about rational autobidders. The platform must know each bidder's private target $t_i$ (or value $v_i$) and budget $B_i$ to solve the allocation LP, and bidders may have incentives to misreport these values. The monotonicity result is therefore about feasibility-preserving allocation under refinement: any feasible coarse LP solution can be lifted to the refined partition while preserving the LP's explicit linear constraints.

### 5.3.4. MAX-CPA NON-MONOTONICITY IN FPA

**Theorem 5.13** (FPA Non-Monotonicity for MAX-CPA Under Designated Profiles). *There exist instances with MAX-CPA bidders in FPA and two feasible uniform-multiplier profiles (one for each model) such that both revenue and welfare decrease under model refinement:*

$$\mathcal{M}_A \succeq \mathcal{M}_B \quad but \quad Rev(\mathcal{M}_A) < Rev(\mathcal{M}_B), \quad (10)$$
$$\mathcal{M}_A \succeq \mathcal{M}_B \quad but \quad Welfare(\mathcal{M}_A) < Welfare(\mathcal{M}_B). \quad (11)$$

Unlike tCPA bidders (who optimally set $\mu = 1$ under FPA without budgets), MAX-CPA bidders strategically shade their bids, and their multipliers can differ across models. This row is not a full equilibrium characterization; it shows that once feasible multiplier profiles are allowed to vary with the information structure, the Jensen monotonicity argument no longer protects either revenue or welfare. Note that with *truly fixed* multipliers across models, revenue would be monotone by the same Jensen argument as Theorem 5.1. See Appendix B.4 for a numerical counterexample demonstrating 66% revenue loss and 0.09% welfare loss under model refinement, together with additive-regret calculations for the displayed profiles.

### 5.3.5. LIQUID WELFARE NON-MONOTONICITY

**Theorem 5.14** (Liquid Welfare Non-Monotonicity). *There exist instances with MAX-CPA bidders having budgets where model refinement decreases liquid welfare:*

$$\mathcal{M}_A \succeq \mathcal{M}_B \quad but$$
$$LiquidWelfare(\mathcal{M}_A) < LiquidWelfare(\mathcal{M}_B). \quad (12)$$

The counterexample in Appendix B.4 demonstrates this: with sufficiently large (non-binding) budgets, liquid welfare equals standard welfare, so the 0.09% welfare decrease applies. More generally, liquid welfare non-monotonicity can also arise via a "crowding out" effect where a budget-constrained high-value bidder displaces an unconstrained bidder.

### 5.4. Discussion

Among strategic auction mechanisms, our results reveal that monotonicity is rare: only FPA with tCPA bidders without budgets guarantees both revenue and welfare monotonicity, while VCG with MAX-CPA bidders without budgets guarantees welfare monotonicity only. Separately, the centralized LP benchmark guarantees welfare monotonicity with budgets, and for tCPA bidders also guarantees surrogate revenue monotonicity. All remaining auction settings studied exhibit non-monotonicity due to: (1) externality-based VCG payments, (2) budget-induced bid shading, or (3) tCPA-VCG misalignment. For platforms, this implies FPA with tCPA autobidders is the safest strategic auction choice, and model improvements should be validated via A/B testing rather than assumed to improve metrics.

## 6. Counterexamples

This section summarizes the numerical counterexamples establishing non-monotonicity. Full calculations appear in Appendix B.

**Representative Counterexample: VCG/SPA Revenue and Welfare Non-Monotonicity.** Two tCPA bidders with $t_A = 10$, $t_B = 1$. Four auctions (one impression each) with conversion probabilities:

|          | Auction 0 | Auction 1 | Auction 2 | Auction 3 |
|----------|-----------|-----------|-----------|-----------|
| Bidder A | 0.20      | 0.05      | 0.01      | 0.01      |
| Bidder B | 0.08      | 0.30      | 0.03      | 0.70      |

Under the *coarse model* (partition $\{\{0,1\}, \{2,3\}\}$), equilibrium multipliers yield: $A$ wins $\{0,1\}$, $B$ wins $\{2,3\}$, with both bidders exactly at their tCPA. Revenue = 3.23, Welfare $= t_A \cdot 0.25 + t_B \cdot 0.73 = 3.23$ (where $0.25 = 0.20+0.05$ and $0.73 = 0.03+0.70$ are total conversions across won auctions).

Under the *fine model* (singletons), equilibrium multipliers shift: $A$ wins only auction 0, $B$ wins $\{1, 2, 3\}$, again with both at tCPA. Revenue = 3.03, Welfare $= t_A \cdot 0.20 + t_B \cdot 1.03 = 3.03$ (where $1.03 = 0.30+0.03+0.70$).

**Result**: Both revenue and welfare decrease by 6.2% despite $\mathcal{M}_A \succeq \mathcal{M}_B$. The intuition: refinement reallocates high-value impressions (auction 1) from the high-$t$ bidder to the low-$t$ bidder, reducing value-weighted conversions. Full calculations in Appendix B.1.

**Key Insights.** VCG/SPA payments depend on competition, which refinement can reduce. Budgets change utilization patterns, potentially eliminating competition in some segments (though LP-based allocation guarantees monotonicity per Theorem 5.11).

# 7. Conclusion

We introduced a formal framework for studying when improvements in prediction models lead to improvements in platform-level metrics in autobidding systems. Our central finding is striking: **model monotonicity is rare.**

**Summary.** We formalized model improvement through a refinement relation inspired by filtrations in probability theory, and systematically analyzed ECM monotonicity across bidder types, auction formats, and budget constraints. Of all settings studied, only **three** guarantee monotonicity: (1) tCPA bidders without budgets under FPA with uniform bidding at $\mu = 1$ (revenue and welfare monotonic); (2) MAX-CPA bidders without budgets under VCG (welfare monotonic only—not revenue); (3) tCPA or MAX-CPA bidders with budgets under LP-based fractional allocation (welfare monotonic; for tCPA, surrogate revenue also monotonic). All remaining settings studied—including VCG/SPA for tCPA bidders and FPA with budgets—exhibit non-monotonicity (verified numerically). For MAX-CPA FPA, we provide a numerical counterexample under designated feasible multiplier profiles demonstrating 66% revenue loss and 0.09% welfare loss; this counterexample also establishes liquid welfare non-monotonicity when using non-binding budgets.

**Incentive Compatibility Trade-off.** Our positive results reveal a fundamental trade-off between monotonicity and incentive compatibility. FPA with uniform bidding is incentive compatible—bidders choose their multipliers privately without revealing targets or budgets to the platform—but only guarantees monotonicity without budget constraints. LP-based allocation handles budgets and guarantees monotonicity, but requires the platform to know private values $(t_i, v_i, B_i)$ to compute the optimal allocation, making it incentive incompatible. No known mechanism achieves both properties simultaneously with budget constraints.

**Implications.** Better models do not always yield better outcomes—platforms should validate improvements through simulation. Budget constraints broadly break monotonicity, and no auction format provides universal guarantees. The interaction between ML models, auctions, and autobidders creates dynamics where component-level improvements may not translate to system-level gains.

# Acknowledgements

The author thanks the anonymous reviewers for their helpful feedback.

# Impact Statement

This paper presents work whose goal is to advance the understanding of interactions between machine learning models, auction mechanisms, and automated bidding systems in online advertising. While our results are theoretical and do not propose a deployed system, they have potential implications for advertising platform design. More informative prediction models can improve allocation quality, but can also enable finer targeting, alter competitive pressure among advertisers, and interact with budget constraints in ways that affect market concentration and access to impressions. These effects may also raise fairness and privacy concerns when refined prediction models rely on sensitive or highly granular user features. Our results therefore support careful system-level validation of model changes rather than evaluating prediction quality in isolation.

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

# A. Proofs of Main Positive Results

This appendix contains complete proofs of the two main positive theorems for the strategic auction settings stated in Section 5 (the LP benchmark proof appears in Appendix B.5). Both proofs rely on Jensen's inequality applied to convex functions (Hardy et al., 1952).

## A.1. Implementation Notes and Additional Remarks

**pCTR/pCVR Implementation.** In practice, platforms may predict CTR and CVR separately: $\text{pCTR}_{\mathcal{M}}(u) := \frac{1}{|C|} \sum_{u' \in C} \text{ctr}(u')$ and $\text{pCVR}_{\mathcal{M}}(u) := \frac{1}{|C|} \sum_{u' \in C} \text{cvr}(u')$. However, our analysis uses only the cluster-averaged conversion probability $p_{i,C}$. The product of cluster-averaged CTR and CVR does not generally equal $p_{i,C}$.

**Per-User vs. Per-Cluster Notation.** ECM definitions sum over users $u \in \mathcal{U}$. In proofs, we equivalently write metrics as weighted sums over clusters: $\text{Revenue}(\mathcal{M}) = \sum_{C \in \mathcal{P}} w_C \cdot \text{Rev}(C)$ where $w_C = |C|/|\mathcal{U}|$. Since all users in a cluster receive identical bids and allocations, per-user sums reduce to per-cluster weighted sums.

**Tie-Breaking.** Ties are broken by advertiser index (lowest index wins). This deterministic rule ensures well-defined allocations and is standard in mechanism design (Krishna, 2009).

## A.2. Proof of Theorem 5.1 (FPA Revenue Monotonicity for tCPA)

*Proof.* Consider tCPA bidders without budget constraints using uniform bidding with optimal multiplier $\mu_i = 1$. Each bidder $i$ bids:

$$b_i(C) = t_i \cdot p_{i,C} \tag{13}$$

where $t_i$ is the target CPA and $p_{i,C}$ is the predicted conversion probability in cluster $C$.

In a first-price auction, the winner pays their bid, so revenue from cluster $C$ is:

$$\text{Rev}(C) = \max_{i \in \mathcal{A}} b_i(C) = \max_{i \in \mathcal{A}} t_i \cdot p_{i,C}. \tag{14}$$

Total expected revenue under model $\mathcal{M}$ with partition $\mathcal{P}$ is:

$$\text{Rev}(\mathcal{M}) = \sum_{C \in \mathcal{P}} w_C \cdot \max_{i \in \mathcal{A}} t_i \cdot p_{i,C}. \tag{15}$$

Now suppose $\mathcal{M}_A \succeq \mathcal{M}_B$, meaning the partition $\mathcal{P}_A$ refines $\mathcal{P}_B$. For each cluster $C^B \in \mathcal{P}_B$, there exist sub-clusters $\{C_1^A, \ldots, C_k^A\} \subseteq \mathcal{P}_A$ that partition $C^B$. By the definition of refinement and calibration:

$$\sum_{j=1}^{k} w_{C_j^A} = w_{C^B}, \tag{16}$$

$$\sum_{j=1}^{k} w_{C_j^A} \cdot p_{i,C_j^A} = w_{C^B} \cdot p_{i,C^B} \quad \forall i \in \mathcal{A}. \tag{17}$$

Define $\lambda_j = w_{C_j^A}/w_{C^B}$, so $\sum_j \lambda_j = 1$ and $\sum_j \lambda_j p_{i,C_j^A} = p_{i,C^B}$.

Define the function $f : \mathbb{R}^n \to \mathbb{R}$ by $f(p_1, \ldots, p_n) = \max_i t_i \cdot p_i$. This function is convex as the pointwise maximum of linear functions.

By Jensen's inequality applied to the convex function $f$:

$$\sum_{j=1}^{k} \lambda_j \cdot f(p_{1,C_j^A}, \ldots, p_{n,C_j^A}) \geq f\left(\sum_{j=1}^{k} \lambda_j p_{1,C_j^A}, \ldots, \sum_{j=1}^{k} \lambda_j p_{n,C_j^A}\right) \tag{18}$$

$$= f(p_{1,C^B}, \ldots, p_{n,C^B}) = \max_i t_i \cdot p_{i,C^B}. \tag{19}$$

Therefore:

$$\sum_{j=1}^{k} w_{C_j^A} \cdot \max_i t_i \cdot p_{i,C_j^A} \geq w_{C^B} \cdot \max_i t_i \cdot p_{i,C^B}. \tag{20}$$

Summing over all clusters $C^B \in \mathcal{P}_B$:

$$\mathrm{Rev}(\mathcal{M}_A) \geq \mathrm{Rev}(\mathcal{M}_B). \tag{21}$$

$\square$

### A.3. Proof of Theorem 5.5 (VCG Welfare Monotonicity for MAX-CPA)

*Proof.* Under VCG, the mechanism allocates to maximize welfare. In each cluster $C$, the allocation is:

$$i^*(C) = \arg\max_{i \in \mathcal{A}} v_i \cdot p_{i,C} \tag{22}$$

where $v_i$ is bidder $i$'s value per conversion. The welfare from cluster $C$ is:

$$W(C) = \max_{i \in \mathcal{A}} v_i \cdot p_{i,C}. \tag{23}$$

Total welfare under model $\mathcal{M}$ with partition $\mathcal{P}$ is:

$$\mathrm{Welfare}(\mathcal{M}) = \sum_{C \in \mathcal{P}} w_C \cdot \max_i v_i \cdot p_{i,C}. \tag{24}$$

This expression has identical structure to FPA revenue in Theorem 5.1, with $v_i$ replacing $t_i$. The function $g(p) = \max_i v_i \cdot p_i$ is convex (pointwise maximum of linear functions).

By the identical Jensen's inequality argument as in the proof of Theorem 5.1:

$$\mathrm{Welfare}(\mathcal{M}_A) \geq \mathrm{Welfare}(\mathcal{M}_B) \tag{25}$$

whenever $\mathcal{M}_A \succeq \mathcal{M}_B$. $\square$

## B. Counterexample Details

This appendix provides detailed calculations for the counterexamples in Section 6.

### B.1. VCG/SPA Revenue and Welfare Non-Monotonicity for tCPA

For single-item auctions, VCG and SPA are equivalent mechanisms (see Remark 4.7). We provide a counterexample demonstrating that *both* revenue and welfare can decrease under refinement.

**Setup:** Four auctions $u \in \{0, 1, 2, 3\}$ (one impression each). Two tCPA bidders $A, B$ with targets $t_A = 10$, $t_B = 1$. Uniform tCPA bidding: each bidder chooses a scalar goal $g_j$ and bids $b_{u,j} = g_j \cdot \hat{p}_{u,j}$ where $\hat{p}$ is the platform's prediction.

**True conversion probabilities:**

|          | Auction 0 | Auction 1 | Auction 2 | Auction 3 |
|----------|-----------|-----------|-----------|-----------|
| Bidder $A$ | 0.20      | 0.05      | 0.01      | 0.01      |
| Bidder $B$ | 0.08      | 0.30      | 0.03      | 0.70      |

**Model B (Coarse):** Partition $\{\{0, 1\}, \{2, 3\}\}$. Predictions are cluster averages:

$$\text{Cluster } \{0, 1\}: \quad \hat{p}_A = (0.20 + 0.05)/2 = 0.125, \quad \hat{p}_B = (0.08 + 0.30)/2 = 0.19 \tag{26}$$

$$\text{Cluster } \{2, 3\}: \quad \hat{p}_A = (0.01 + 0.01)/2 = 0.01, \quad \hat{p}_B = (0.03 + 0.70)/2 = 0.365 \tag{27}$$

**Equilibrium (both at tCPA):** Bidders choose uniform multipliers to maximize expected conversions subject to expected CPA $\leq t_i$, taking others' multipliers as given. In SPA, any multipliers in an interval preserve the allocation; we exhibit an equilibrium with both CPA constraints binding to make the welfare and revenue calculations transparent. The following multipliers constitute such an equilibrium: $A$ wins $\{0, 1\}$, $B$ wins $\{2, 3\}$:

$$g_B = \frac{t_A \cdot (p_{0,A} + p_{1,A})}{\hat{p}_{0,B} + \hat{p}_{1,B}} = \frac{10 \times 0.25}{0.38} = 6.579 \tag{28}$$

$$g_A = \frac{t_B \cdot (p_{2,B} + p_{3,B})}{\hat{p}_{2,A} + \hat{p}_{3,A}} = \frac{1 \times 0.73}{0.02} = 36.5 \tag{29}$$

**Best-response verification:** Each bidder's multiplier is set so their CPA constraint binds exactly. Increasing the multiplier would cause the bidder to win additional auctions where their cost-per-conversion exceeds their target, violating feasibility. For each bidder, the best-response *set* includes all multipliers that preserve the same winning set while satisfying the CPA constraint; among these, we select the CPA-binding multiplier as a canonical equilibrium. Other equilibria with the same allocation exist within these intervals.

**Verification:** Bids in $\{0, 1\}$: $b_A = 36.5 \times 0.125 = 4.56$, $b_B = 6.579 \times 0.19 = 1.25$ ($A$ wins). Bids in $\{2, 3\}$: $b_A = 0.365$, $b_B = 2.40$ ($B$ wins). Both CPAs bind: $A$ pays 2.50 for 0.25 conversions (CPA $= 10 = t_A$); $B$ pays 0.73 for 0.73 conversions (CPA $= 1 = t_B$).

$$\text{Rev}(\mathcal{M}_B) = 2.50 + 0.73 = 3.23 \tag{30}$$

$$\text{Welfare}(\mathcal{M}_B) = t_A \cdot 0.25 + t_B \cdot 0.73 = 2.50 + 0.73 = 3.23 \tag{31}$$

**Model A (Fine):** Partition into singletons $\{\{0\}, \{1\}, \{2\}, \{3\}\}$, so $\hat{p}_{i,j} = p_{i,j}$.

**Equilibrium (both at tCPA):** Allocation: $A$ wins only $\{0\}$, $B$ wins $\{1, 2, 3\}$. From CPA constraints:

$$g_B = \frac{t_A \cdot p_{0,A}}{\hat{p}_{0,B}} = \frac{10 \times 0.20}{0.08} = 25 \tag{32}$$

$$g_A = \frac{t_B \cdot (p_{1,B} + p_{2,B} + p_{3,B})}{\hat{p}_{1,A} + \hat{p}_{2,A} + \hat{p}_{3,A}} = \frac{1 \times 1.03}{0.07} = 14.71 \tag{33}$$

**Best-response verification:** As in the coarse model, each bidder's CPA constraint binds exactly. Increasing either multiplier would cause that bidder to win additional auctions violating their CPA target. Best-response sets are intervals preserving the winning set; we select CPA-binding multipliers as canonical. The same allocation persists across all equilibria in these intervals.

**Verification:** Auction 0: $b_A = 2.94$, $b_B = 2.0$ ($A$ wins). Auctions 1,2,3: $B$ wins (e.g., auction 1: $b_A = 0.74$, $b_B = 7.5$). CPAs bind: $A$ pays 2.0 for 0.20 conversions (CPA $= 10$); $B$ pays 1.03 for 1.03 conversions (CPA $= 1$).

$$\text{Rev}(\mathcal{M}_A) = 2.0 + 1.03 = 3.03 \tag{34}$$

$$\text{Welfare}(\mathcal{M}_A) = t_A \cdot 0.20 + t_B \cdot 1.03 = 2.0 + 1.03 = 3.03 \tag{35}$$

**Non-monotonicity:** Both revenue and welfare decrease: $3.23 \to 3.03$ (6.2% loss). Both bidders win $\geq 1$ auction and are exactly at tCPA in both conditions. Welfare decreases because refinement reallocates auction 1 (where $A$ has high value $t_A p_{1,A} = 0.5$ vs $B$'s value $t_B p_{1,B} = 0.3$) from high-$t$ bidder $A$ to low-$t$ bidder $B$.

## B.2. FPA Revenue Non-Monotonicity with Budgets

For the FPA counterexample with budget constraints (Theorem 5.10):

**Setup:** Four auctions, two advertisers. First-price auction with uniform bidding: $b_i(u) = \alpha_i \cdot \hat{p}_{i,u}$.

- Advertiser 1: Budget $B_1 = 3.185$, target $t_1 = 8.674$ (tCPA non-binding; budget binds under these multipliers).

- Advertiser 2: Budget $B_2 = \infty$, target $t_2 = 1.662$.

**Budget model:** The budget $B_i$ is a deterministic ex ante constraint on total expected payment across all auctions: $\sum_{u \in \mathcal{U}} x_i(u) \cdot b_i(u) \leq B_i$, where $x_i(u) \in \{0, 1\}$ indicates whether advertiser $i$ wins auction $u$. No pacing or randomization is used; bidders commit to a single multiplier $\alpha_i$ applied uniformly. The budget-binding multiplier is the maximum $\alpha_i$ such that expected spend equals $B_i$ given the allocation induced by $(\alpha_1, \alpha_2)$.

**True conversion probabilities:**

| Auction | $q_1$ | $q_2$ |
|---------|-------|-------|
| 1 | 0.516 | 0.559 |
| 2 | 0.027 | 0.850 |
| 3 | 0.560 | 0.617 |
| 4 | 0.555 | 0.330 |

**Model B (Coarse):** Partition $\{1, 2\}, \{3, 4\}$.

$$\text{Predictions:} \quad \hat{p}_{1,\{1,2\}} = 0.2715, \quad \hat{p}_{1,\{3,4\}} = 0.5575 \tag{36}$$
$$\hat{p}_{2,\{1,2\}} = 0.7045, \quad \hat{p}_{2,\{3,4\}} = 0.4735 \tag{37}$$

Candidate multiplier profile: $\alpha_1 = 2.8565$ (budget binding), $\alpha_2 = t_2 = 1.662$.

Advertiser 1's bids: $b_1(\{1, 2\}) = 0.7755, b_1(\{3, 4\}) = 1.5925$.
Advertiser 2's bids: $b_2(\{1, 2\}) = 1.1709, b_2(\{3, 4\}) = 0.7870$.

Winners: Advertiser 2 wins auctions 1,2; Advertiser 1 wins auctions 3,4.
Revenue: $\text{Rev}(\mathcal{M}_B) = 2 \times 1.1709 + 2 \times 1.5925 = 5.5268$.

**Model A (Fine):** Singleton partition (predictions equal true probabilities).

Candidate multiplier profile: $\alpha_1 = 1.9528$ (budget binding), $\alpha_2 = t_2 = 1.662$.

Advertiser 1's bids: $b_1(1) = 1.0076, b_1(2) = 0.0527, b_1(3) = 1.0936, b_1(4) = 1.0838$.
Advertiser 2's bids: $b_2(1) = 0.9291, b_2(2) = 1.4127, b_2(3) = 1.0255, b_2(4) = 0.5485$.

Winners: Advertiser 1 wins auctions 1,3,4; Advertiser 2 wins auction 2.
Revenue: $\text{Rev}(\mathcal{M}_A) = 1.0076 + 1.4127 + 1.0936 + 1.0838 = 4.5977$.

**Profile justification and best-response analysis:** In both models, Advertiser 1's budget binds, determining their multiplier by budget exhaustion. In Model B, this yields $\alpha_1 = 2.8565$ with spend $= B_1 = 3.185$. Given this $\alpha_1$, Advertiser 2's best-response *set* is an interval: all $\alpha_2 \in (1.10, \infty)$ preserve winning $\{1, 2\}$ (threshold: $\alpha_2 > b_1(\{1, 2\})/\hat{p}_{2,\{1,2\}} \approx 1.10$). We define a canonical equilibrium selection that picks $\alpha_2 = t_2 = 1.662$—the conversion-maximizing choice that exactly binds the tCPA constraint. For Advertiser 1, we define the canonical selection as the maximal multiplier that exhausts their budget ($\alpha_1 = 2.8565$). The profile $(2.8565, 1.662)$ is one Nash equilibrium selected from the continuum of equilibria within the best-response intervals; the same selection convention applies to Model A with $\alpha_1 = 1.9528$.

**Result:** Revenue decreases by $(5.5268 - 4.5977)/5.5268 = 16.8\%$.

**Mechanism:** Under the coarse model, Advertiser 1's low predicted probability in cluster $\{1, 2\}$ means they lose these auctions despite high $\alpha_1$. Under the fine model, Advertiser 1 can win auction 1 (high true probability) with a lower $\alpha_1$. The budget-constrained advertiser wins more auctions but pays less per auction, while the unconstrained advertiser loses a high-payment auction (auction 1). Net effect: total revenue decreases.

**Liquid welfare calculation:** Liquid welfare is $\sum_i \min(B_i, \text{welfare}_i)$ where $\text{welfare}_i = \sum_{u \in W_i} t_i \cdot p_{i,u}$ (Advertiser $i$'s total value from won impressions $W_i$).

*Model B:* Advertiser 1 wins $\{3, 4\}$: welfare $= 8.674 \times (0.560 + 0.555) = 9.67$, capped at $\min(3.185, 9.67) = 3.185$. Advertiser 2 wins $\{1, 2\}$: welfare $= 1.662 \times (0.559 + 0.850) = 2.34$. Total: $3.185 + 2.34 = 5.53$.

*Model A:* Advertiser 1 wins $\{1, 3, 4\}$: welfare $= 8.674 \times (0.516 + 0.560 + 0.555) = 14.15$, capped at $3.185$. Advertiser 2 wins $\{2\}$: welfare $= 1.662 \times 0.850 = 1.41$. Total: $3.185 + 1.41 = 4.60$.

Liquid welfare decreases by $(5.53 - 4.60)/5.53 = 16.8\%$, matching revenue. The decrease arises because model refinement

lets the budget-constrained advertiser win auction 1 (displacing the unconstrained advertiser), but the budget cap prevents extracting additional value while the unconstrained advertiser loses conversions.

### B.3. VCG Revenue Non-Monotonicity for MAX-CPA

For the VCG counterexample with MAX-CPA bidders (Theorem 5.9):

**Setup:** $v_1 = 10$, $v_2 = 8$. Segments with $w_1 = w_2 = 0.5$:

|          | $S_1$ | $S_2$ |
|----------|-------|-------|
| Bidder 1 | 0.2   | 0.6   |
| Bidder 2 | 0.5   | 0.1   |

**Model B (Coarse):**

$$v_1 p_1 = 10 \times 0.4 = 4.0, \quad v_2 p_2 = 8 \times 0.3 = 2.4 \tag{38}$$

$$\text{VCG payment} = 2.4, \quad \text{Welfare} = 4.0 \tag{39}$$

**Model A (Fine):**

$$S_1: \quad v_1 p_1 = 2.0, \quad v_2 p_2 = 4.0 \quad \Rightarrow \text{Bidder 2 wins, payment} = 2.0 \tag{40}$$

$$S_2: \quad v_1 p_1 = 6.0, \quad v_2 p_2 = 0.8 \quad \Rightarrow \text{Bidder 1 wins, payment} = 0.8 \tag{41}$$

**Results:**

$$\text{Rev}(\mathcal{M}_A) = 0.5 \times 2.0 + 0.5 \times 0.8 = 1.4 < 2.4 = \text{Rev}(\mathcal{M}_B) \tag{42}$$

$$\text{Welfare}(\mathcal{M}_A) = 0.5 \times 4.0 + 0.5 \times 6.0 = 5.0 > 4.0 = \text{Welfare}(\mathcal{M}_B) \tag{43}$$

This confirms that VCG welfare is monotonic for MAX-CPA (as proven in Theorem 5.5), but revenue is non-monotonic. The revenue decrease is 41.7%.

### B.4. FPA Non-Monotonicity for MAX-CPA

We provide a numerical counterexample demonstrating that FPA with MAX-CPA bidders can exhibit non-monotonicity in both revenue and welfare under model-dependent feasible multiplier profiles (Theorem 5.13). This is not a full equilibrium characterization; we report additive-regret calculations below to show that the profiles are near best responses in this instance.

**Setup:** Two impression types $H$ (high-value) and $L$ (low-value) with probabilities $\Pr(H) = 0.9$ and $\Pr(L) = 0.1$. Two MAX-CPA bidders:

| Advertiser | Value $v_i$ | $p_i(H)$ | $p_i(L)$ |
|------------|-------------|----------|----------|
| 1          | 600         | 0.2      | 0.01     |
| 2          | 10          | 0.02     | 0.5      |

**Model B (Coarse):** The coarse model pools both impression types. Calibrated predictions:

$$\hat{p}_1 = 0.9 \times 0.2 + 0.1 \times 0.01 = 0.181 \tag{44}$$

$$\hat{p}_2 = 0.9 \times 0.02 + 0.1 \times 0.5 = 0.068 \tag{45}$$

Expected values per impression: $v_1 \hat{p}_1 = 108.6$, $v_2 \hat{p}_2 = 0.68$.

Under uniform bidding, consider multipliers $(g_1, g_2) = (0.68/0.181, 10) \approx (3.7569, 10)$. Bidder 1 bids $b_1 = (0.68/0.181) \times 0.181 = 0.68$ and Bidder 2 bids $b_2 = 10 \times 0.068 = 0.68$. With ties broken by lowest index, Bidder 1 wins all impressions.

**Revenue:** $\text{Rev}(\mathcal{M}_B) = 0.68$.

**Welfare:** Bidder 1 captures all value: $\text{Welfare}(\mathcal{M}_B) = v_1 \hat{p}_1 = 108.6$.

**Model A (Fine):** The fine model distinguishes $H$ and $L$ impressions.

In segment $H$: $v_1 p_1(H) = 600 \times 0.2 = 120$, $v_2 p_2(H) = 10 \times 0.02 = 0.2$.
In segment $L$: $v_1 p_1(L) = 600 \times 0.01 = 6$, $v_2 p_2(L) = 10 \times 0.5 = 5$.

Under the fine model, consider the multiplier profile $(g_1, g_2) = (1, 1)$.

**Profile justification:** With $(g_1, g_2) = (1, 1)$: Bidder 1's bid in $H$ is $0.2 > 0.02$ (Bidder 2's bid), so Bidder 1 wins $H$; Bidder 2's bid in $L$ is $0.5 > 0.01$ (Bidder 1's bid), so Bidder 2 wins $L$. Other multiplier profiles also preserve this allocation whenever the ratio satisfies $0.1 < g_1/g_2 < 50$. As stated in Assumption 4.10, MAX-CPA FPA counterexamples use designated feasible multipliers without full equilibrium claims; non-monotonicity arises because different information structures can induce different shading profiles.

With these multipliers:

Segment $H$: $b_1(H) = 1 \times 0.2 = 0.2$, $b_2(H) = 1 \times 0.02 = 0.02$. Bidder 1 wins.
Segment $L$: $b_1(L) = 1 \times 0.01 = 0.01$, $b_2(L) = 1 \times 0.5 = 0.5$. Bidder 2 wins.

**Revenue:**

$$\text{Rev}(\mathcal{M}_A) = 0.9 \times 0.2 + 0.1 \times 0.5 = 0.18 + 0.05 = 0.23 \tag{46}$$

**Welfare:**

$$\text{Welfare}(\mathcal{M}_A) = 0.9 \times 120 + 0.1 \times 5 = 108 + 0.5 = 108.5 \tag{47}$$

**Results:**

$$\text{Revenue decrease:} \quad \frac{0.68 - 0.23}{0.68} = 66.2\% \tag{48}$$

$$\text{Welfare decrease:} \quad \frac{108.6 - 108.5}{108.6} = 0.09\% \tag{49}$$

**Additive-regret calculation.** In the coarse model, $(g_1, g_2) = (0.68/0.181, 10)$ is a threshold best-response profile under the stated tie-breaking rule: Bidder 1 wins all impressions at the lowest bid that ties Bidder 2, and Bidder 2 would obtain non-positive surplus by outbidding. In the fine model, $(g_1, g_2) = (1, 1)$ is not an exact Nash equilibrium. However, its additive best-response regrets are small in the scale of the instance. Bidder 1's utility is $0.9(120 - 0.2) = 107.82$; by lowering $g_1$ to just above 0.1, Bidder 1 still wins $H$ and obtains utility approaching $0.9(120 - 0.02) = 107.982$, a gain of at most 0.162. Bidder 2's utility is $0.1(5 - 0.5) = 0.45$; by lowering $g_2$ to just above 0.02, Bidder 2 still wins $L$ and obtains utility approaching $0.1(5 - 0.01) = 0.499$, a gain of at most 0.049. Thus the fine profile is an approximate best-response profile, not an exact equilibrium.

**Mechanism:** The key insight is that model refinement enables *market segmentation* that reduces competitive pressure. Under the coarse model, Bidder 1 faces competition across all impressions and must bid at the pooled threshold to win. Under the fine model, each segment becomes a near-monopoly in bid space: Bidder 1 dominates $H$ (bid 0.2 vs. 0.02) and Bidder 2 dominates $L$ (bid 0.5 vs. 0.01). This segmentation allows both bidders to shade their bids more ($g_1 = g_2 = 1$), dramatically reducing payments. The welfare loss is minimal because the fine model still allocates efficiently within each segment; the small loss arises from Bidder 2 winning segment $L$ where Bidder 1 has slightly higher value (6 vs. 5).

**B.5. LP Benchmark Monotonicity Proof**

This appendix provides the complete proof of Theorem 5.11, showing that a centralized LP allocation-and-payment benchmark guarantees revenue and welfare monotonicity under model refinement. This is not an equilibrium claim about strategic autobidders; the LP directly enforces the relevant feasibility constraints.

**LP for tCPA Bidders:** For tCPA bidders, the LP maximizes total welfare subject to a surrogate-payment budget:

$$\max_{x_{i,C} \geq 0} \quad \sum_{i \in \mathcal{A}} \sum_{C \in \mathcal{P}} w_C \cdot x_{i,C} \cdot t_i \cdot p_{i,C} \qquad \text{(maximize total welfare)} \tag{50}$$

$$\text{s.t.} \quad \sum_{C \in \mathcal{P}} w_C \cdot x_{i,C} \cdot t_i \cdot p_{i,C} \leq B_i \qquad \forall i \in \mathcal{A} \text{ (budget constraint: spend} \leq B_i) \tag{51}$$

$$\sum_{i \in \mathcal{A}} x_{i,C} \leq 1 \qquad \forall C \in \mathcal{P} \text{ (supply constraint)} \tag{52}$$

$$x_{i,C} \in [0,1] \qquad \forall i \in \mathcal{A}, C \in \mathcal{P} \tag{53}$$

Here, the LP's surrogate payment for bidder $i$ in cluster $C$ equals $t_i \cdot p_{i,C}$ per impression, matching truthful tCPA first-price payments at $\mu_i = 1$.

**LP for MAX-CPA Bidders:** For MAX-CPA bidders, the LP maximizes welfare subject to a value-budget/liquid-welfare feasibility constraint:

$$\max_{x_{i,C} \geq 0} \quad \sum_{i \in \mathcal{A}} \sum_{C \in \mathcal{P}} w_C \cdot x_{i,C} \cdot v_i \cdot p_{i,C} \qquad \text{(maximize total welfare)} \tag{54}$$

$$\text{s.t.} \quad \sum_{C \in \mathcal{P}} w_C \cdot x_{i,C} \cdot v_i \cdot p_{i,C} \leq B_i \qquad \forall i \in \mathcal{A} \text{ (budget constraint)} \tag{55}$$

$$\sum_{i \in \mathcal{A}} x_{i,C} \leq 1 \qquad \forall C \in \mathcal{P} \text{ (supply constraint)} \tag{56}$$

$$x_{i,C} \in [0,1] \qquad \forall i \in \mathcal{A}, C \in \mathcal{P} \tag{57}$$

**Remark (Budget semantics):** The LP result should be read as a centralized allocation-and-payment benchmark. It explicitly enforces feasibility constraints of the calibrated linear form $\sum_C w_C x_{i,C} a_i p_{i,C} \leq B_i$, where $a_i$ is fixed for bidder $i$. For tCPA bidders at $\mu_i = 1$, the surrogate payment $t_i p_{i,C}$ coincides with value per impression. For MAX-CPA bidders, the LP welfare result uses the analogous value-budget/liquid-welfare constraint $v_i p_{i,C}$. Strategic auction spend budgets, where actual payments depend on the auction format and other bidders' bids, are handled separately in the FPA and VCG rows of Table 1.

*Proof of Theorem 5.11.* Let $x^B = \{x^B_{i,C_B}\}$ be an optimal solution for the coarse model $\mathcal{M}_B$.

Construct a feasible solution $x^A$ for the refined model $\mathcal{M}_A$ by copying each coarse allocation fraction to every sub-cluster: for each coarse cluster $C_B \in \mathcal{P}_B$ with refined sub-clusters $\{C_1^A, \ldots, C_k^A\}$, define

$$x^A_{i,C_j^A} := x^B_{i,C_B} \quad \text{for all } i \in \mathcal{A} \text{ and } j = 1, \ldots, k. \tag{58}$$

**(1) Supply feasibility.** For any refined cluster $C_j^A \subseteq C_B$:

$$\sum_i x^A_{i,C_j^A} = \sum_i x^B_{i,C_B} \leq 1, \tag{59}$$

so all refined supply constraints hold.

**(2) Budget feasibility.** For any advertiser $i$:

$$\sum_{C \in \mathcal{P}_A} w_C \cdot x^A_{i,C} \cdot t_i \cdot p_{i,C} = \sum_{C_B \in \mathcal{P}_B} \sum_{j=1}^{k} w_{C_j^A} \cdot x^B_{i,C_B} \cdot t_i \cdot p_{i,C_j^A} \tag{60}$$

$$= \sum_{C_B \in \mathcal{P}_B} x^B_{i,C_B} \cdot t_i \cdot \left( \sum_{j=1}^{k} w_{C_j^A} \cdot p_{i,C_j^A} \right). \tag{61}$$

By calibration preservation, $\sum_{j=1}^{k} w_{C_j^A} \cdot p_{i,C_j^A} = w_{C_B} \cdot p_{i,C_B}$. Thus:

$$= \sum_{C_B \in \mathcal{P}_B} x_{i,C_B}^B \cdot t_i \cdot w_{C_B} \cdot p_{i,C_B} = \sum_{C_B \in \mathcal{P}_B} w_{C_B} \cdot x_{i,C_B}^B \cdot t_i \cdot p_{i,C_B} \leq B_i. \tag{62}$$

All budget constraints hold. (The argument is identical for MAX-CPA with $v_i$ replacing $t_i$.)

**(3) Objective preservation.** The same calculation shows the refined solution achieves exactly the same objective value:

$$\sum_{C \in \mathcal{P}_A} \sum_i w_C \cdot x_{i,C}^A \cdot t_i \cdot p_{i,C} = \sum_{C_B \in \mathcal{P}_B} \sum_i w_{C_B} \cdot x_{i,C_B}^B \cdot t_i \cdot p_{i,C_B}. \tag{63}$$

Since $x^A$ is feasible for $\mathcal{M}_A$, the refined optimum cannot be smaller.

**tCPA case:** With objective $\sum w_C x_{i,C} t_i p_{i,C}$:

$$\mathrm{Rev}_{\mathrm{LP}}(\mathcal{M}_A) \geq \mathrm{Rev}_{\mathrm{LP}}(\mathcal{M}_B). \tag{64}$$

**MAX-CPA case:** With objective $\sum w_C x_{i,C} v_i p_{i,C}$ (welfare):

$$\mathrm{Welfare}_{\mathrm{LP}}(\mathcal{M}_A) \geq \mathrm{Welfare}_{\mathrm{LP}}(\mathcal{M}_B). \tag{65}$$

The algebra is identical in both cases, substituting $v_i$ for $t_i$. $\square$

### B.6. Notes on Liquid Welfare

**Liquid Welfare:** The counterexample in Appendix B.4 establishes liquid welfare non-monotonicity: with sufficiently large (non-binding) budgets, liquid welfare equals standard welfare, so the 0.09% welfare decrease applies directly. More generally, non-monotonicity can also arise via a "crowding out" effect where a budget-constrained high-value bidder displaces an unconstrained bidder, and the constrained bidder's capped contribution is less than the displaced bidder's full contribution.

## C. Open Problems

Our work suggests several directions for future research:

1. *Approximate monotonicity*—can we bound the magnitude of monotonicity violations?

2. *Incentive-compatible monotonic mechanisms*—do mechanisms exist that achieve both properties?

3. *Calibration and misspecification*—how do the positive and negative results change when prediction models are imperfectly calibrated or only approximately refine one another?

4. *Multi-slot auctions*—which monotonicity results extend to multi-slot formats such as generalized second-price or generalized first-price auctions?

5. *Equilibrium selection*—can stronger equilibrium-based characterizations be obtained for first-price auctions with MAX-CPA bidders, where pure equilibria and selection can be delicate?

6. *Empirical prevalence*—how often do the non-monotonic effects identified by the counterexamples arise in simulations or deployed advertising systems?

7. *Learning dynamics*—how do model changes affect autobidder learning and convergence?

8. *Heterogeneous populations*—what happens with mixed tCPA/MAX-CPA bidder populations?

