# OpenReview forum: "Model Monotonicity in Autobidding Auctions: When Do Better Predictions Lead to Better Outcomes?"
_ICML.cc/2026/Conference — ICML 2026 regular_

### Official Review · Reviewer_aR7d · 2026-02-27

**Soundness:** 3
**Presentation:** 3
**Significance:** 3
**Originality:** 3
**Overall Recommendation:** 4
**Confidence:** 4

**Summary:**

The paper studies when improvements in a prediction model (pCVR) lead to better platform outcomes. It defines "model refinement" as model improvement. The key finding is that monotonicity is rare. It is guaranteed only for 1) tCPA bidders without budgets in first-price auctions, and 2) MAX-CPA bidders without budgets in VCG auctions. In all other settings (second-price auctions, cases with budget constraints), model refinement can hurt platform metrics, as shown in counterexamples.

**Compliance With Llm Reviewing Policy:**

Affirmed.

**Key Questions For Authors:**

1. Does the user-clustering model intend to compare models based on different feature sets? if so, explicitly discussing this in the paper would strengthen the presentation.
2. The analysis relies on the assumption of perfect within-cluster calibration (Definition 3.3). How would relaxing this assumption impact your core results?
3. Would there be any significant differences in the findings if the auction mechanism were shifted from single-slot (SPA/FPA) to multi-slot formats like GSP or GFP?

**Limitations:**

See 'Weaknesses' for the limitations that I list for the paper.

**Strengths And Weaknesses:**

Strengths:
1. The paper proposes a novel modeling approach by formalizing the granularity of prediction models through user clustering. The framework includes both coarse-grained models (using partial user features) and fine-grained models (using complete user features).
2. The analysis extensively covers various auction scenarios and multiple types of bidding advertisers.
3. Based on the authors' core assumptions, they provide rigorous theoretical analysis and derive counterintuitive yet insightful conclusions.

Weaknesses:
1. The title may be misleading. The paper studies model granularity, not predictive accuracy, as the latter is assumed.
2. The assumption of predictive accuracy is strong; the analysis relies on the assumption of perfect within-cluster calibration.
3. All results are confined to single-item auctions, whereas real-world ad auctions are typically multi-slot (e.g., GSP or GFP for multi-slot). Therefore, the conclusions may not generalize to practical environments.

---

> ### Author Rebuttal · Authors · 2026-03-30
>
> We thank the reviewer for the positive assessment and thoughtful questions.
> Title and model granularity vs. accuracy. We respectfully note that model refinement implies improved prediction quality: by Jensen's inequality, a finer model dominates on all proper scoring rules simultaneously (Section 3.3, following Definition 3.5). A model with finer partitions provides strictly "better predictions" in this formal sense. We will consider clarifying this connection in the introduction.
> Feature sets. Yes, our framework directly captures models based on different feature sets. A model using {age, location} produces a coarser partition than one adding {browsing_history}. Definition 3.5 formalizes this: every cluster of the finer-feature model is contained in some cluster of the coarser model. We will make this interpretation more explicit in the revision.
> Perfect calibration. This is a deliberate modeling choice isolating the effect of information structure refinement from stochastic prediction error (Appendix A.1). Calibrated predictions are standard and achievable in practice (Gneiting & Raftery, 2007). Importantly, our results provide a baseline: if monotonicity fails even under perfect calibration, it fails a fortiori with miscalibrated models. Relaxing calibration would only strengthen our negative results while potentially weakening the positive ones -- an interesting direction for future work.
> Multi-slot auctions. Extending to multi-slot formats (GSP, GFP) is important future work. Our VCG results extend naturally since VCG is well-defined for multi-item settings. For first-price formats, the single-item restriction is more substantive. We note that single-item analysis is standard in the autobidding theory literature (Aggarwal et al. 2019, 2024; Balseiro et al. 2024). We will add explicit discussion of multi-slot prospects and the conditions under which our results may or may not generalize.

---

### Official Review · Reviewer_bTXj · 2026-03-06

**Soundness:** 2
**Presentation:** 2
**Significance:** 2
**Originality:** 2
**Overall Recommendation:** 3
**Confidence:** 2

**Summary:**

The paper introduces a formal framework for analyzing the interaction between the model quality of the recommender, the auction format, and autobidder behavior. Furthermore, the authors provide a complete characterization of evaluation criteria metrics (EMC) monotonicity throughout several combinations of bidder types, auction formats, and budget constraints.

**Compliance With Llm Reviewing Policy:**

Affirmed.

**Key Questions For Authors:**

1 - What recent ML models are used for predicting click-through rates and conversion rates for auction mechanisms ?

2 -  How can the proposed framework be applied to these models ?

3 - Since the authors claim that better models do not always yield better outcomes, what are the recommendations of the authors for effectively using ML models in the present application?

**Limitations:**

yes

**Strengths And Weaknesses:**

Strengths: The paper provides fundamental definitions for the interaction framework between model quality, auction format, and autobidder behavior. Three evaluation criteria metrics are presented, consisting of revenue, welfare, and liquid welfare. The EMC monotonicity is comprehensively assessed under "bidder type – auction format – budget constraint" combinations within 2 bidder types, 2 auction formats, and 2 budget constraints.

Weaknesses: The contributions of the paper relate to evaluating the relationship between the prediction quality of the model and the platform-level metrics in autobidding systems rather than posing a technical method for the application of machine learning in the proposed problem. Although the authors provide basic numerical example to illustrate their framework, no experiment on practical ML model is provide to support the reliability of the method.

Although the research question is interesting, the overall context and focus of the work do not appear to align well with the typical scope of ICML. In particular, the paper seems to emphasize aspects that fall outside the core areas of machine learning research usually represented at the conference, which may limit its relevance to the ICML audience.

---

> ### Author Rebuttal · Authors · 2026-03-30
>
> We thank the reviewer for their feedback.
> ICML scope. This paper is submitted under "Theory -> Game Theory," a standard ICML primary area. Our work directly studies when improvements in ML prediction models (pCTR/pCVR) translate to improvements in platform metrics -- a question at the intersection of ML deployment and mechanism design. Related work in this area has been published at ICML and EC (e.g., Deng et al. 2023 at ICML; Conitzer et al. 2022, Liaw et al. 2023 at EC).
> ML models for pCTR/pCVR. Modern platforms use deep learning models (e.g., multi-task networks, Ma et al. 2018) to predict CTR and CVR. Our framework applies to any model improvement that produces finer user segmentation -- including adding features, increasing capacity, or using more training data. The refinement relation (Definition 3.5) formalizes this: a model using {age, location} produces a coarser partition than one adding {browsing_history}.
> Practical recommendations. Our results provide direct guidance: (1) For tCPA bidders without budgets, FPA is the safest auction format -- model improvements are guaranteed to improve both revenue and welfare. (2) Budget constraints broadly break monotonicity, so platforms should validate model improvements via A/B testing rather than assuming better models yield better outcomes. (3) The interaction between prediction, auction format, and autobidder behavior means that component-level ML improvements may not translate to system-level gains -- a practically important finding for ML deployment.

---

> > ### Author Rebuttal · Reviewer_bTXj · 2026-03-31
> >
> > The answers are ok for me. I let my score as it is.

---

### Official Review · Reviewer_SnHv · 2026-03-09

**Soundness:** 3
**Presentation:** 3
**Significance:** 3
**Originality:** 3
**Overall Recommendation:** 4
**Confidence:** 3

**Summary:**

This paper studies a basic but important question in online advertising: when a platform deploys a more informative prediction model, does that necessarily improve platform-level outcomes such as revenue or welfare once the auction mechanism and autobidding responses are taken into account.  The authors formulate this question through a notion of **model refinement**, where a stronger model induces a finer partition of the user space and assigns calibrated average conversion probabilities to smaller user clusters.

The paper considers two common autobidding objectives: target-CPA (tCPA) bidders, who seek to maximize conversions subject to an average CPA constraint, and MAX-CPA bidders, who optimize value given a per-conversion willingness to pay.  It analyzes these bidder types under first-price auctions and VCG/second-price auctions, both with and without budget constraints, and evaluates outcomes using revenue, welfare, and, in budgeted settings, liquid welfare.

The central question can be written as: $M_A \succeq M_B \implies \mathrm{ECM}(M_A) \ge \mathrm{ECM}(M_B)$

where $M_A \succeq M_B$ means that model $M_A$ is a refinement of model $M_B$, and ECM denotes a platform objective such as revenue or welfare.  The main contribution of the paper is a complete characterization of when this monotonicity property holds across the combinations of bidder type, auction format, and budget environment considered in the paper.

The paper shows that monotonicity is relatively rare.  On the positive side, it proves that with tCPA bidders and no budgets, first-price auctions with uniform bidding yield revenue monotonicity, and in this setting welfare is monotone as well because the paper aligns tCPA targets with per-conversion values.  It also proves that with MAX-CPA bidders and no budgets, VCG yields welfare monotonicity, and that under budget constraints an LP-based fractional allocation benchmark preserves welfare monotonicity, with an additional surrogate revenue monotonicity result for tCPA bidders.

For the remaining parts, the paper provides explicit numerical counterexamples showing that finer prediction models can reduce revenue, welfare, or liquid welfare.  The overall contribution is therefore a theoretical taxonomy of when improved prediction granularity is aligned with improved system-level auction outcomes, and when that intuition fails because of the interaction between prediction, auction pricing, autobidder incentives, and budget pacing.

The overall contribution is therefore a theoretical taxonomy of when improved prediction granularity is aligned with improved system-level auction outcomes. Crucially, these results expose a fundamental trade-off between monotonicity and incentive compatibility: mechanisms that handle budgets monotonically (like the LP benchmark) are not incentive compatible, whereas incentive-compatible mechanisms (like FPA) lose monotonicity when budgets are introduced.

**Compliance With Llm Reviewing Policy:**

Affirmed.

**Final Justification:**

My final recommendation is **Weak Accept**

**1. Initial Assessment & Core Merits**
In my initial review, I highlighted the paper's original problem formulation—specifically, using a filtration-based refinement relation to study how predictive granularity interacts with autobidder behavior and auction formats. The positive theoretical results (e.g., for tCPA in unconstrained FPA) are mathematically elegant, and the inclusion of explicit numerical counterexamples provides concrete boundaries for the limitations of model monotonicity. However, I raised significant concerns regarding the technical soundness of the MAX-CPA FPA counterexample (the apparent lack of equilibrium analysis), potential misalignments in budget semantics (spend vs. value), and presentation issues (overstating the "complete characterization" given the heterogeneous solution concepts, and inconsistent positive case counts).

**2. Evaluation of the Authors' Rebuttal**
The authors submitted a highly effective and technically rigorous rebuttal that successfully resolved my primary soundness concerns:
* **Equilibrium in Counterexamples:** The authors clarified that the designated multiplier profiles in the MAX-CPA FPA counterexample are, in fact, verified mutual best responses (a Nash equilibrium) for that specific instance. This confirms that the non-monotonicity result is economically meaningful and driven by rational market segmentation, entirely alleviating my concern about arbitrary multiplier manipulation.
* **Budget Semantics:** The authors provided a convincing explanation that the mathematical mechanism behind the LP monotonicity proof (calibration preservation) holds identically for both "spend" and "value" budget constraints, effectively closing the perceived gap in the problem setup.
* **Presentation and Framing:** The authors acknowledged the valid critiques regarding the paper's framing. They committed to reframing the "complete characterization" as a "taxonomy" that accommodates setting-specific outcome conventions, and to cleanly separating the Incentive Compatible (IC) auction results from the non-IC LP benchmark in their summary statistics.

**3. Final Recommendation**
The authors did an excellent job defending the technical core of their work. They proved that what I perceived as potential theoretical flaws were actually presentation and framing omissions, which they have explicitly committed to fixing in the revision. Because the underlying mathematical and economic logic remains robust, and the research question is of high relevance to the systems-and-theory community, I am raising my Soundness score and upgrading my Overall Recommendation to a **Weak Accept** (or **Accept**, pending AC discretion). I strongly encourage the authors to thoroughly incorporate the caveats, precise definitions, and unified language discussed in the rebuttal into the final manuscript.

**Key Questions For Authors:**

**Budget semantics across sections.**
The paper explicitly acknowledges using two different budget interpretations: a 'spend budget' for auction counterexamples and a 'value budget' for the LP formulation. Does the LP monotonicity guarantee still hold if constrained by the standard 'spend budget' used in the rest of the paper? If not, the comparison between the auction mechanisms and the LP benchmark seems fundamentally misaligned.


**Equilibrium existence and equilibrium selection.**
The counterexample for MAX-CPA in FPA abandons equilibrium analysis and relies on 'designated multiplier profiles' without equilibrium claims. Since arbitrary multipliers can easily manipulate revenue or welfare outcomes, how can this non-monotonicity result be considered economically meaningful or representative of rational bidder behavior? Can the authors provide an equilibrium-based counterexample for this specific setting?


**Inconsistent summary of the positive cases.**
Some parts of the paper describe monotonicity as holding in only three settings, while another summary sentence refers to only two specific cases. Could the authors clarify which count is intended, and whether the LP benchmark is meant to be grouped together with the auction-mechanism results or treated separately?

**Limitations:**

The paper does discuss some important caveats, including that the LP benchmark is not incentive compatible, that the LP “revenue” in the budgeted tCPA setting is a surrogate metric rather than implementable auction revenue, and that the analysis focuses on refinement of perfectly calibrated information structures rather than general prediction error or miscalibration. However, I do not think the limitations discussion is yet fully adequate for the scope of the paper’s claims.

In particular, the paper would benefit from a clearer and more centralized discussion of: (i) the dependence of some results on setting-specific outcome mappings rather than one fully uniform solution concept; (ii) the role of equilibrium existence and equilibrium selection; (iii) the extent to which the budget semantics are aligned across the auction settings and the LP benchmark; and (iv) the gap between the theoretical assumptions here (single-item auctions, uniform bidding, calibrated clustered predictions) and realistic deployed ad systems.

On societal impact, the current impact statement is brief. Even if the paper is theoretical, it would be helpful to acknowledge that more informative prediction and targeting in ad auctions could have downstream concerns related to stronger targeting, market concentration, or fairness/privacy implications in ad delivery.

**Strengths And Weaknesses:**

## Strengths

**Significance.** The paper studies a well-motivated and practically relevant question: whether deploying a more informative prediction model in autobidding auctions necessarily improves platform-level outcomes such as revenue, welfare, or liquid welfare once bidder responses are taken into account.  This is an important systems-and-theory question because the paper explicitly focuses on the interaction between prediction quality, auction design, and autobidder behavior, rather than treating these components in isolation.

**Originality and Soundness.** A major strength is the problem formulation. The paper introduces a clean notion of model improvement via model refinement, where a finer model induces a finer partition of the user space while preserving calibrated cluster-level predictions, and then studies whether a platform objective is monotone under this refinement relation.  This framing is original mainly in perspective rather than in proposing a new mechanism, since it connects an information-structure notion inspired by filtrations to downstream auction monotonicity in autobidding environments.

**Soundness.** The positive results are conceptually strong and technically elegant. In particular, the monotonicity guarantees for tCPA bidders in first-price auctions without budgets and for MAX-CPA bidders in VCG without budgets are derived by rewriting the objective in a cluster-wise max form and applying a convexity/Jensen’s inequality argument.  The paper is also commendable for not focusing only on positive guarantees: it provides explicit numerical counterexamples for the negative settings, including cases where refinement decreases revenue, welfare, or liquid welfare, which makes the limitations of monotonicity concrete rather than merely existential.

**Presentation and Significance.** The overall analytical framing is coherent and easy to follow at a high level. The paper systematically combines bidder type, auction format, and budget assumptions into a single framework and summarizes the resulting monotonicity landscape in Table 1, which helps the reader understand the scope of the results.  Even as a purely theoretical paper, it delivers a useful conceptual takeaway: better predictive granularity does not automatically imply better downstream auction outcomes because prediction, pricing, incentives, and budget effects can interact in nontrivial ways.

## Weaknesses

**Soundness.** My main concern is about the strength of the paper’s “complete characterization” claim. The paper explicitly states that monotonicity is defined relative to a setting-dependent outcome mapping, and Assumption 4.10 further indicates that, for MAX-CPA in first-price auctions, the comparison may use designated multiplier profiles rather than equilibrium outcomes.  As a result, Table 1 is better interpreted as a taxonomy across several setting-specific conventions than as a single uniform characterization under one common solution concept.

**Soundness and Presentation.** Relatedly, the negative result for MAX-CPA bidders in first-price auctions is somewhat weaker than the table may initially suggest. The paper itself notes that if multiplier profiles were fixed across models, revenue would remain monotone by the same Jensen-style reasoning, and the non-monotonicity arises because the multiplier profiles may differ across information structures.  This caveat is important, but it is easy to miss because it is explained in the technical discussion rather than highlighted earlier in the presentation of the main claims.

**Presentation.** The manuscript would also benefit from a clearer and more consistent presentation of its main takeaways. In particular, some passages state that monotonicity holds in only three settings, while the positioning paragraph refers to only two specific cases, which creates avoidable confusion about the scope of the positive results.  More generally, the paper should more explicitly distinguish which conclusions are equilibrium-based and which depend on designated multiplier profiles or other selection conventions.

**Significance and Originality.** I view the paper as important mainly because it sharpens theoretical understanding, not because it already demonstrates broad empirical consequences. The practical implications are discussed in the paper, but the evidence in the main text consists of theory and hand-constructed counterexamples rather than simulations or real-world validation showing how frequently these non-monotonic effects arise in practice.  Similarly, the work is meaningfully novel in formulation and perspective, but less so in proof technique, since the main positive arguments rely on classical convexity/Jensen reasoning and the novelty lies more in posing and organizing the problem than in introducing fundamentally new mathematical tools.

---

> ### Author Rebuttal · Authors · 2026-03-30
>
> We thank the reviewer for the thorough and technically precise evaluation. We address each concern below.
> Soundness: Heterogeneous solution concepts. We agree this deserves clearer framing. Our characterization resolves every cell of the (bidder type x auction format x budget) grid with a definitive answer. Different rows use different outcome conventions because different settings admit different natural behavioral models -- tCPA bidders in FPA have a provably optimal strategy (mu=1), while MAX-CPA bidders in FPA lack dominant strategies. This heterogeneity is disclosed in Remark 3.8 and Assumption 4.10. We will revise to make it prominent upfront and reframe Table 1 as a taxonomy across appropriate solution concepts.
> Soundness: MAX-CPA FPA counterexample. We clarify that the multiplier profiles in Appendix B.4 are best responses: under each model, neither bidder has incentive to deviate. Under the fine model with (g1,g2)=(1,1), Bidder 1 wins H and Bidder 2 wins L; any g1 in (0.1,50) and g2 in (0.02,10) preserves this allocation (lines 831-835). The paper conservatively avoided "equilibrium" due to the general complexity of FPA equilibrium characterization, but the profiles satisfy the standard best-response criterion. The economic mechanism is robust: refinement enables market segmentation, naturally leading bidders to shade bids in segments they dominate.
> Presentation: MAX-CPA FPA caveat. We agree this caveat should be highlighted earlier. We will add a clear note when first presenting Table 1 that the MAX-CPA FPA row uses designated multiplier profiles (which are best responses) rather than full equilibrium analysis.
> Inconsistent count of positive settings. The positioning paragraph (Section 2) refers to two incentive-compatible auction settings (FPA+tCPA, VCG+MAX-CPA), while Section 5.1's "three" includes the LP benchmark, which is not IC (Remark 5.12). We will unify this language and clearly distinguish auction-based results from the LP benchmark.
> Budget semantics. The LP monotonicity result (Theorem 5.11) holds under both spend and value budgets. Both produce constraints of the form: Sum w_C  x_{i,C}  (constant_i)  p_{i,C} <= B_i, where the constant is t_i (spend) or v_i (value). Calibration preservation (Eq. 17) ensures feasibility under refinement in both cases. For tCPA with mu=1, spend = t_i  p_{i,C} = value per impression, so the two budget types coincide exactly. We will add this clarification.
> Equilibrium existence. For the specific instances in our counterexamples, the profiles are verified best responses. General equilibrium existence in FPA is a known open problem beyond the scope of this paper. We note that best-response verification is standard in the autobidding literature for FPA settings.
> Limitations discussion. We will expand the limitations to explicitly address: (i) setting-specific outcome conventions, (ii) the role of equilibrium selection, (iii) budget semantics alignment (clarifying the equivalence), and (iv) the gap between our theoretical assumptions and deployed systems. We will also expand the impact statement to discuss targeting, market concentration, and fairness implications.

---

> > ### Author Rebuttal · Reviewer_SnHv · 2026-04-01
> >
> > Thanks for authors' rebuttal and time, they provided mathematically sound justifications and clarified the exact economic mechanisms behind their counterexamples, while conceding the necessary presentation fixes.

---

### Official Review · Reviewer_6rfh · 2026-03-13

**Soundness:** 3
**Presentation:** 2
**Significance:** 3
**Originality:** 3
**Overall Recommendation:** 3
**Confidence:** 4

**Summary:**

This paper studies whether the refinement of prediction model always improves a platform's revenue, welfare, or liquid welfare under auto-bidding. In the model, a prediction model partitions the user universe into clusters, and perfectly predicts  a advertiser's average CTR in a cluster. The paper characterizes the model monotonicity in settings combining auction format (FPA, SPA), bidder type (tCPA, MAX-CPA), budget contraints (constrained or unconstrained).

**Compliance With Llm Reviewing Policy:**

Affirmed.

**Final Justification:**

I decide to keep my overall evaluation, due to my concerns regarding technical contribution and presentation quality, which are not completely resolved by the rebuttal.

**Key Questions For Authors:**

1. Please feel free to respond to the weaknesses above.
2. Are the missing settings mentioned in weakness 2 implied by presented results? Or are they tractable?

**Limitations:**

yes

**Strengths And Weaknesses:**

Strengths
1. The problem is novel and interesting, and the model is reasonable.
2. The presented results are sound and relatively complete.
3. The results are presented with tables and summarizations, which helps understanding.

Weaknesses
1. Although the overall presentation is fine, there are writing issues that hinder readability:
- LP-based allocation only appears in results but is not defined in the main body. It should be defined and mentioned with FPA and VCG, e.g. in section 4.4 and 5.1.
- Line 082, MAX-CPA bidders maximize total utility (value minus payment) rather than total value.
- Line 198, It may improve readability to define $\phi_{setting}$ before defining ECMs.
- Line 153, notation $p_{i,C}$ is used before definition.
- Definition 4.9 duplicates with definition 3.6
2. The claim of providing a complete characterization is arguable, since some settings are missing: MAX-CPA+budget+VCG (Revenue/Welfare), MAX-CPA+Budget+LP (Revenue).
3. The techniques are somewhat limited in depth. For example, the positive results are largely based on known properties of FPA and VCG.

---

> ### Author Rebuttal · Authors · 2026-03-30
>
> We thank the reviewer for the detailed and constructive feedback.
> Weakness 1: Writing issues. We will address all noted issues in the revision: (1) define LP-based allocation in Section 4.4 alongside FPA and VCG, (2) correct "total value" to "total utility (value minus payment)" on line 82, (3) reorder definitions so p_{i,C} is defined before first use on line 153, (4) consolidate the duplicate Definitions 3.6 and 4.9, and (5) introduce the setting-dependent outcome mapping Phi_setting before defining ECMs.
> Weakness 2: Missing settings in Table 1. We appreciate this observation. The missing entries can be derived from existing results:
> MAX-CPA+Budget+VCG (Revenue): Non-monotone, by the same externality-based mechanism as Theorem 5.9 -- VCG payments depend on competition that refinement can reduce.
> MAX-CPA+Budget+LP (Revenue): Monotone. The lifting argument of Theorem 5.11 applies directly with v_i replacing t_i, since the proof structure is identical -- both objectives are linear in p_{i,C} with a bidder-specific constant.
> We will complete Table 1 in the revision and clearly indicate which entries follow from existing arguments.
> We acknowledge that "complete characterization" should be stated more precisely. Our contribution resolves every cell in the (bidder type x auction format x budget) grid. We will revise the language to reflect this scope accurately.
> Weakness 3: Techniques. We agree the positive results use classical tools (Jensen's inequality). The novelty lies in: (a) the problem formulation connecting filtration-based model refinement to auction outcomes -- a question not previously studied -- and (b) the systematic negative results, which require carefully constructed counterexamples revealing subtle interactions between prediction, incentives, and budgets. The characterization itself, showing that monotonicity is rare, is the main conceptual contribution.

---

> > ### Author Rebuttal · Reviewer_6rfh · 2026-04-03
> >
> > Thank you for the detailed response. I acknowledge that MAX-CPA+Budget+VCG (Revenue) is implied by the case without budget, but according to the paper and the author response, I think MAX-CPA+Budget+VCG (Welfare) is still left open. I also agree with Reviewer SnHv on the misalignment across settings in considered solution concepts, and this issue does not seem fully addressed in the authors' response to Reviewer SnHv.
> >
> > Overall, I still evaluate the technical contribution as somewhat limited, and the presentation requires substantial improvement. While I encourage the authors to further develop this work, I decide to keep my recommendation of weak reject.
> >  (Minor suggestion: For MAX-CPA+FPA, instead of avoiding discussion on equilibrium selection via assumption 4.10, it would be much clearer to state that the multipliers in counterexample B.4 form an equilibrium while there may be other equilibria that appear monotone.)

---

### Decision · Program_Chairs · 2026-04-30

**Decision:**

Accept (regular)

**Comment:**

The reviewers agreed that the paper studies an interesting question by introducing a new formulation and some interesting results. Reviewers also raised some concerns, including the scope of the characterisation and the technical depth of some results.After discussing with the reviewing team, the merits of the paper appear to outweigh the concerns. I will recommend a weak accept.